# Decision-margin consistency: a principled metric for human and machine performance alignment

**George A. Alvarez**
Department of Psychology
Kempner Institute
Harvard University
Cambridge, MA 02138
alvarez@wjh.havard.edu

**Talia Konkle**
Department of Psychology
Center for Brain Science & Kempner Institute
Harvard University
Cambridge, MA 02138
talia_konkle@harvard.edu

## Abstract

Understanding the alignment between human and machine perceptual decision-making is a fundamental challenge. While most current vision deep neural networks are deterministic and produce consistent outputs for the same input, human perceptual decisions are notoriously noisy [1]. This noise can originate from perceptual encoding, decision processes, or even attentional fluctuations, leading to different responses for the same stimulus across trials. Thus, any meaningful comparison between human-to-human or human-to-machine decisions must take this internal noise into account to avoid underestimating alignment. In this paper, we introduce the **decision-margin consistency metric**, which draws on signal detection theory, by incorporating both the variability in decision difficulty across items and the noise in human responses. By focusing on decision-margin distances—-continuous measures of signal strength underlying binary outcomes—-our method can be applied to both model and human systems to capture the nuanced agreement in item-level difficulty. Applying this metric to existing visual categorization datasets reveals a dramatic increase in human-human agreement relative to the standard error consistency metric. Further, human-to-machine agreement showed only a modest increase, highlighting an even larger representational gap between these systems on these challenging perceptual decisions. Broadly, this work underscores the importance of accounting for internal noise when comparing human and machine error patterns, and offers a new principled metric for measuring representational alignment for biological and artificial systems.

## 1 Introduction

One of the central goals in neuroAI is to understand the degree to which machine and human decision-making align, particularly in visual tasks. This goal is crucial because, as machines increasingly take on roles in critical areas like medical diagnosis, autonomous driving, and robotics, their ability to replicate human decision patterns can directly influence performance, trust, and safety. Quantifying alignment between machine and human decision-making is thus an important endeavor. Here, we focus on perceptual decision making, considering patterns of performance on classification tasks among vision models and human observers.

Geirhos et al. (2020) addressed this challenge by introducing the metric of trial-by-trial *error consistency* [2]. They highlight that while two systems might achieve the same level of accuracy on a task, they might arrive at this overall level by making completely different mistakes, pointing to differences in their underlying decision processes. In their study, they examined a set of models trained on the Imagenet dataset, and measured their visual categorization accuracy in a series of probe

experiments where the visual images had been manipulated, including sihouettes, line-drawings, and texture-shape conflict images. Naturally, these models make many more errors on these out-of-distribution images–and, a benefit of being in this intermediate performance regime is that we have access to the patterns of errors these models make over images. The logic is that systems which make consistent errors with each other are more likely to be processing visual information in similar ways. They ran a similar categorization test in humans, using shorter presentation time with masks to ensure the human behavioral responses were off-ceiling and that errors are made. Then, using their error-consistency metric, they found that while models were similar to other models and humans to other humans in the errors they made, there was a significant gap between humans and models. And, they found human perceptual decisions were much more inconsistent which each other than models to models. This metric has since become a standard in assessing human-machine representational alignment in the field.

However, an important limitation of the error consistency method and single-trial experimental paradigm is that it does not account for noise in human decision-making. Unlike current deterministic computer vision systems, humans do not always give the same exact response to the same exact stimulus even when performing the same exact task (much to the chagrin of psychophysicists)— a phenomenon often attributed to random fluctuations throughout the biological system which are internal to the person and independent of the stimulus [3–5]. When comparing errors between machines and humans to make inferences about their representational alignment, we need metrics that isolate the component of human errors driven by perceptual encoding constraints, separate from the stimulus-independent noise contaminating human responses.

To address this need, we draw on insights from signal detection theory (SDT; [6–10]). Instead of focusing on alignment of binary outcomes (correct or incorrect), decisions can instead be characterized in both humans and machines by *decision-margin distance*, which is a continuous measure of signal strength underlying a binary choice. The decision margin distance is an estimate of the internal activation strength for the correct option, relative to largest internal activation amongst incorrect options. Intuitively, you can think of the decision margin as the strength of evidence in favor of the target category over non-target categories. The logic here is that some classifications are intrinsically easier or harder due to underlying perceptual encoding constraints, and that two systems with similar perceptual-decision algorithms should agree on the relative difficulty of each decision.

In this paper, we introduce a **decision-margin consistency** metric, computed as the Pearson correlation between the decision margin-distances across all items for two sets of decision makers (e.g., human-vs-human, or model-vs-model, or model-vs-human). This metric takes into account both the noise in human decisions and the graded variability in decision difficulty across items, to provide a principled picture of how aligned decision-makers are. Note that decision-margin consistency changes the target from single-trial responses (which are noisy in humans) to relative item-by-item difficulty (pearson correlation over decision margin distances), which can be estimated with much higher statistical reliability. Applying this metric to existing datasets substantially increases estimates of human-human agreement, and also strongly increases machine-machine agreement. While human-machine agreement increases as well, the improvement is far less than the increase in agreement between humans, suggesting that the gulf between human and machine decisions is greater than estimated without taking internal noise into account.

Our novel contribution is not to develop a new measure of human decision-making ability — instead we draw on decades of research in human psychophysics which has developed both a strong theoretical framework (*Signal Detection Theory*), and performance metrics that disentangle different factors that influence human decisions, such as perceptual signal strength versus internal noise or response bias [11, 7]. Our contribution is to show that the same signal-detection framework can be applied both to ANN decision makers and human decision-makers, and that the decision-margin-consistency metric is a principled and statistically powerful metric for comparing item-by-item decisions for any pair of decision makers (e.g., human vs. human, machine vs. machine, human vs. machine). Based on this framework and approach, we offer recommendations for optimizing the design of human experiments to take internal noise into account and to ensure highly powered evaluation tasks. Code and data are available at: `https://github.com/harvard-visionlab/decision-margin-consistency`, and `https://github.com/harvard-visionlab/model-vs-human-dmc`

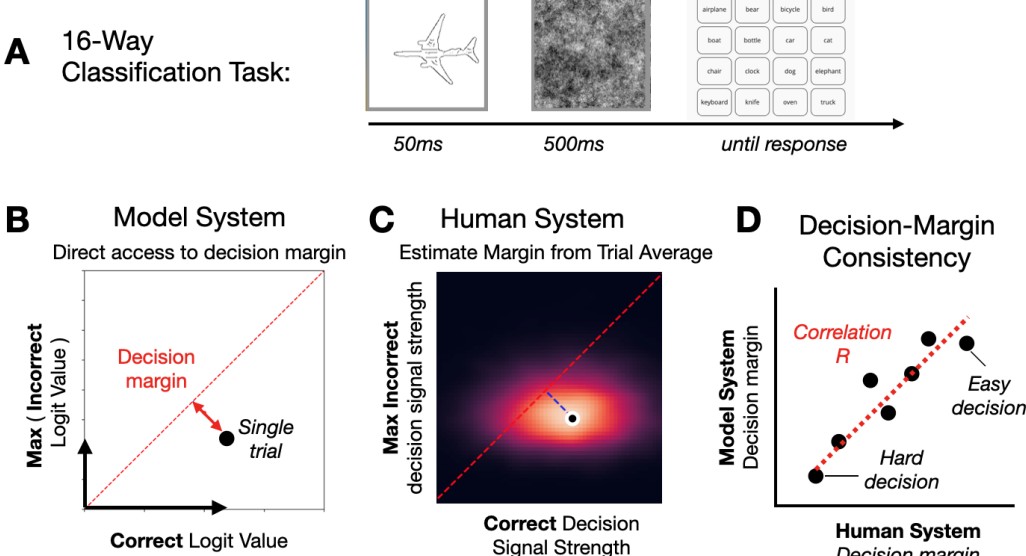

Figure 1: **A.**Perceptual decision task adapted from Geirhos et al., 2020. **B,C,D**. Signal Detection Framework for decision making. For model system(**B**), the decision margin can be directly calculated from the correct logit value and the maximum of the incorrect logit values. For human systems **(C)**, any single trial can be construed as a sample from an underlying distribution of both perceptual evidence and noise for each response option, where the final decision is taken as the max of these options. **(D)** After computing and/or estimating the decision margin for each item, the decision margin consistency between any two systems is computed as the correlation over these item decision strengths.

## 2 Background: Human Decisions are Noisy

In human psychophysics and decision-making research, it is taken as a given that humans and other animals respond variably from trial-to-trial [1], even when performing the exact same task on the exact same stimulus. Moreover, even when decision-makers do make the same choice on exact trial repetitions, they tend to take considerably variable amounts of time to arrive at that decision, implying a noisy information accumulation process underlying their decisions [12]. Here we demonstrate the impact of this noise on human decisions.

Throughout this section we report means and 95% confidence intervals calculated by computing the standard error of the mean (SEM) and using the Student's $t$-distribution to account for the sample size, estimating the range within which the true population mean likely falls.

### 2.1 Human Error Consistency

Consider the task introduced by Geirhos et al., [13] in which participants perform a 16-alternative forced choice (16-AFC) visual task (see **Figure 1A** for our slightly modified version of their task). Participants are briefly shown a target image followed by a mask to prevent extended processing, and then asked to choose the correct category from 16 possible options. Geirhos et al., [2] computed the error consistency between two decision-makers on this task using Cohen's Kappa, which quantifies the degree of agreement between sets of decisions, adjusting for chance agreement (which depends on the overall accuracy of each observer):

$$\kappa_{i,j} = \frac{c_{\text{obs}_{i,j}} - c_{\text{exp}_{i,j}}}{1 - c_{\text{exp}_{i,j}}} \tag{1}$$

Where $c_{\text{obs}i,j}$ is the observed error consistency between decisions of two decision makers $(i, j)$, and $c_{\text{exp}i,j}$ is the consistency expected to happen just by chance, given the overall percent correct for each decision maker:

$$c_{\text{obs}i,j} = \frac{1}{N} \sum_{k=1}^{N} \mathbb{I}\left[y_k^{(i)} = y_k^{(j)}\right] \qquad (2)$$

where $N$ is the total number of decisions and $y_k$ is the correctness (0,1) of the $kth$ decision, and

$$c_{\text{exp}_{i,j}} = p_i p_j + (1 - p_i)(1 - p_j) \qquad (3)$$

where $p_i$ is the overall proportion correct for decision maker $i$, and $p_j$ is the overall proportion correct for decision maker $j$.

Their general finding with this measure was that human decision makers are not particularly consistent with each other. For example, in the case of the decisions over edge-filtered images, the average overall accuracy of human participants was at a useful regime of 87.1% correct, but their average error-consistency measured using Cohen's kappa which adjusts for chance agreement, was only 0.32 (95% CI: [0.28, 0.36]).

From these values, it may be tempting to infer that individual human visual systems are quite different from each other in how they process information, since the trials in which they are accurate and inaccurate are so inconsistent. Indeed, this would be warranted if one were comparing one deterministic model to another. However, this inference assumes that the estimate of each individual's response on a given trial is stable. But, just how consistent are these human perceptual decisions?

## 2.2 Self-consistency on single-trials decisions is low

How consistent are people with themselves when they perform the exact same trial multiple times? In order to quantify this self-consistency, we replicated the edge-image classification experiment with one addition: participants saw each image two different times, rather than only once (see **Appendix A** for experiment details). Note that overall accuracy across participants was 80% (95% CI: [77%, 82%]), avoiding both ceiling and floor performance, which is a reasonable regime for error comparison. Item level accuracy scores (N=160), averaged across participants (N=45), spanned from near chance (1/16 = 6.25%) to 100%, and were skewed towards the higher end (most items between 50% and 100% accuracy), indicating that item-level difficulty spanned the full range.

We found that a person's *self error-consistency* between their own first and second response was only $k = 0.54$ (95% CI: [0.50, 0.57]). This value is only slightly higher than the between-person consistency (mean=0.31, 95% CI: [0.30, 0.32]) computed based on the first-trial for each item. In other words, an individual is not very self-consistent on a trial-by-trial basis. When we simply correlate the first and second responses of an individual, this yields a similar result, but helps put the noise level into perspective: Participants responses on trial 1 correlate with their responses on trial 2 at $r = 0.55$, or $(r^2 = 0.31)$, indicating that up to 69% of the variance in single trial responses in this task is attributable to noise.

If these individual trial responses are corrupted by random, independent noise, then that noise should cancel out when averaging across trials. Note that, when we *average across trials*, we are in effect shifting the analysis goal to estimate how *difficult a given item is* for that individual decision-maker (e.g., here averaging across two trials, yielding scores of 1, 0.5, or 0 depending on each trial's accuracy). On average across all pairs of people, we found that the item-by-item correlation between any two people increases from $r = .33$ (single response per item) to $r = 0.43$ (average of two trials per item; 95% CI: [0.42, 0.44]). This result shows that while individual decisions are noisy, this noise cancels out when averaging, revealing systematic information about item difficulty. In principle, these benefits of averaging would accrue if we had more trials to average — to the extent that the response variability is due to stochastic noise.

In short, this small experiment demonstrates that error consistency measures of individual human decisions, as defined on a trial-by-trial basis, are quite noise-limited, and are underestimating the true level of agreement between human perceptual decisions.

## 2.3 Challenges for estimating per-item difficulty in a single human decision maker

In this section, we outline why it is challenging to accurately estimate the difficulty of any single item for a particular decision maker, and we describe how we can address this challenge by averaging trials (both within and across different human participants), and by focusing on relative difficulty across items (Pearson correlation of difficulty scores across items) rather than the absolute difficulty of any single item.

A key question is, how many trials are needed to accurately estimate item-level difficulty for a *single human decision-maker*? Simulations reveal a stark reality. For example, consider a single image with a ground-truth accuracy of 75% correct for a particular individual. In this case, you would need to collect 300 trials (repetitions of the same image), just for the 95% confidence interval to be within +/- 5% of the true accuracy for that image (see **Appendix B** for simulation details).

These estimates make no assumptions about noise, and only reflect what to expect based on sampling error alone. To provide an intuition, consider the number of heads expected when tossing a fair coin N times. For small Ns, the proportion of heads will frequently deviate substantially from 50% (e.g., with N=1, it will never be 50%). Likewise, if the true probability of a correct response for a given item is 50%, there will be error in estimating the true probability correct from a sample of N trials. The numbers reported above (and **Appendix B**) are based on simulating the impact of sampling error alone — taking other factors into account can only increase the number of trials required. Thus, establishing reliable estimates of per-item accuracy in individual subjects for a reasonably large set of items is a daunting challenge, and is the reason the field of perceptual psychophyics focused on small-N (number of subjects) designs with sometimes thousands of trials per participant [14].

One solution to this challenge is to focus on estimating the relative difficulty across items (i.e., the pearson correlation across individual item scores), rather than the absolute difficulty of any single item. Moreover, rather than estimating the item-level difficulty for each individual human participant, it is also possible and principled to average data across trials from different participants to estimate item-level difficulty in the population. In our simulations (**Appendix B.1**, we find that 20-40 participants are needed to obtain highly consistent estimates of item-level difficulty (e.g., correlation > 90% between groups of 40 participants). Empirically, this approach yields reliable group-level estimates of item-level decision difficulty across a large number of items in the current dataset: computing the average across subjects, the average percent correct per item for first responses correlate highly with the average percent correct for second responses, at a remarkable $r = 0.937$ ($r^2$=.88; in other words, 88% of the variance is signal!). Beyond these statistical power benefits, this approach is also justified theoretically, drawing on extensive work in signal detection theory (detailed further below), as a way to estimate the underlying signal-strength driving decisions in the context of noise. Finally, a further benefit of this approach is that it can be applied to existing datasets where fewer data points are collected per participant and per item.

One potential issue with using the group average is that the average accuracy may not fully reflect the decisions that any one individual would make. However, it turns out empirically that visual systems tend to be quite similar across people, at least in broad terms (e.g. [15], but see [16]). As a result, the relative difficulty of visual items tends to be systematic and stable across individuals, and can serve as a useful and meaningful target for model-to-human alignment.

## 3 Method

### 3.1 A signal detection framework for modeling perceptual decisions

Signal detection theory (STD) was developed to model human decision making under uncertainty and noise [6–10], but the framework can be applied to both human and machine decision making, with typical deep neural network classifiers being a special case of noise=0. Further, while it is traditionally associated with detection tasks (e.g. detecting the presence of a target among background noise, with hits, false alarms, misses, and correct rejections), the framework is also applicable to n-alternative-forced choice tasks.

A central concept of signal detection theory is signal strength, which feeds into the decision-making process. In an n-alternative forced-choice task, the signal strength for the correct answer is based on the positive evidence for the correct option. Every other option also has a signal strength proportional

to the evidence in favor of each option. Items with relatively more evidence for the correct response than all alternatives are easy to classify, while items with similar evidence for the correct and at least one incorrect option are harder to classify. What ultimately matters is the relative strength for the target option and the maximal response amongst the n-1 incorrect options. This can be visualized as distance from the *decision margin* between these options, where signal strength for the correct option is plotted along the x-axes and signal-strength for the maximal non-target option is plotted on the y-axis (**Figure 1**). Items with strong strength on both possible alternatives are near to the decision boundary and systematically difficult, while items that with relatively little evidence for any incorrect alternative options are far from the decision boundary and systematically easy.

### 3.2 Estimating Decision-Margin Distance

In deep neural network models, we can easily compute this decision margin distance for any given item $i$, because we have direct access to the output activations for each of the categories:

$$ModelDM_i = \frac{S_c - \max_{j=1, j \neq c}^{M}(S_j)}{\sqrt{2}} \tag{4}$$

where $S_c$ is the logit activation for the correct category, and $S_j$ is the logit activation for the $j$th incorrect category, among the $M$ possible options. This distance corresponds to the red-double headed arrow indicated in **Figure 1B.**

In humans, we do not have direct access to the internal activation strengths for each decision option, so we cannot directly compute the decision margin. Instead, the standard signal detection approach is estimate the *sensitivity* or signal-to-noise ratio $d'$. Mathematically, in an n-alternative-forced-choice task, the proportion of correct responses (i.e. the average accuracy) is, in fact, the bias-free measure of $d'$ [11], unless observers are strongly biased towards particular responses, in which case additional measures are needed to obtain an unbiased estimate [17]. For simplicity, here we assume no response bias, averaging over repetitions of trials (either within a person, or across people), to compute an average proportion correct measure for the item, which provides a continuous value used to index the decision-margin distance.

To compute the human decision-margin index for each item, we calculated the mean accuracy ($P_i$) by averaging the correctness of all responses to the item:

$$HumanDMI_i = P_i = \frac{1}{n_i} \sum_{j=1}^{n_i} k_{i,j} \tag{5}$$

where $n_i$ is the total number of responses for item $i$, and $k_{i,j}$ is the response correctness for the $j$-th response (0, or 1).

We refer to these mean-proportion correct scores as the human decision-margin-index (DMI) to emphasize that it isn't a direct measure of the decision-margin, but is being used as an indirect index of the decision margin.

### 3.3 Decision-Margin Consistency

Given that we only have an indirect index of the decision-margin distance in humans, we cannot directly compare human-percent correct to model decision-margin scores. Instead, we opt to correlate these scores, which quantifies how the *relative* differences in proportion correct in humans are related to the *relative* differences in decision-margin distance. Thus, once the decision-margin (or the proxy proportion correct) has been computed for each item, for two different systems $A$ and $B$ (either models or humans), the **decision-margin consistency** is calculated simply as the Pearson correlation of these item-level decision margins.

$$DMC_{A,B} = correlation(A_{DM}, B_{DM}) \tag{6}$$

### 3.4 Computing Human-to-Human Decision-Margin Consistency

To estimate the human vs. human decision margin consistency, also known as the "noise-ceiling" of the data, a standard split-half reliability procedure can be used. Participants are split into two even groups, and the mean correct per item is computed separately for each group, over which DMC is computed as in equation (6) above. Note that since only half of the data are entering into the correlation of human-to-human consistency, this will underestimate the consistency of the full dataset; we apply the standard Spearman-Brown prophecy equation to adjust this correlation accordingly [18, 19]:

$$r_{full} = \frac{2 \cdot r_{12}}{1 + r_{12}} \tag{7}$$

where $r_{12}$ is the Pearson correlation between the split-halves. Finally, this procedure was repeated for all possible splits of the data into two halves, and DMC scores were averaged over all splits to compute the final estimate of human vs. human DMC. Confidence intervals can then be estimated empirically with bootstrapping, from this set of all possible splits of the participants.

## 4 Results: Humans vs. Models

We compared *error consistency* and *decision margin consistency* between models and humans, computed over three experiments conducted by Geirhos et al., 2020 [2]. In these experiments, the participant's task was always 16-way classification, while the probe image set was varied, and included silhouetted objects, edged stimuli, and cue-conflict stimuli.

For the set of models under consideration, we curated a set of off-the-shelf pre-trained models, spanning different architectural classes, all trained on the ImageNet dataset. For the human data, we re-analyzed these publicly available datasets. We then computed both error consistency and decision margin consistency scores for all human-to-human, DNN-to-DNN, and model-to-human comparisons.

The results are shown in **Figure 2**. Both error consistency and Decision-Margin Consistency are plotted for each DNN as a function of 16-way classification accuracy. The horizontal reference lines indicate the average human-to-human consistency (magenta), DNN-to-DNN consistency (tan), humans, and the DNN-to-human consistency (gray). The shaded area depicts the 95% confidence interval, reflecting true individual variability for DNN vs. DNNs (which are noiseless), and true individual variability plus noise in humans.

There are three major observations that hold across all experiments:

1. Human-to-human consistency increases substantially when using decision margin consistency. Indeed the degree of consistency is now even higher than DNN-to-DNN consistency, which is a qualitatively different pattern of results than observed with error-consistency. This result reveals that humans' perceptual decision making processes are indeed highly related, even though single trials are quite noisy.

2. Model-to-model consistency is higher when evaluated with the decision margin metric, compared to the error consistency metric. How can this be, given that these are noiseless systems? The logic here is that error consistency can only relate binary 0's and 1's, whereas the decision-margin metric gives a more parametric measure of how hard or easy a decision is, which gives more dynamic range for strong relationships between models.

3. DNN-to-Human consistency, on average across these models, also shows a slight increase when comparing the decision-margin relationships. However, the gap between the other two reference lines is, if anything, larger; indicating that these distinctive models are still much more like each other than they are with the average human response when it comes to consistent item-level decisions.

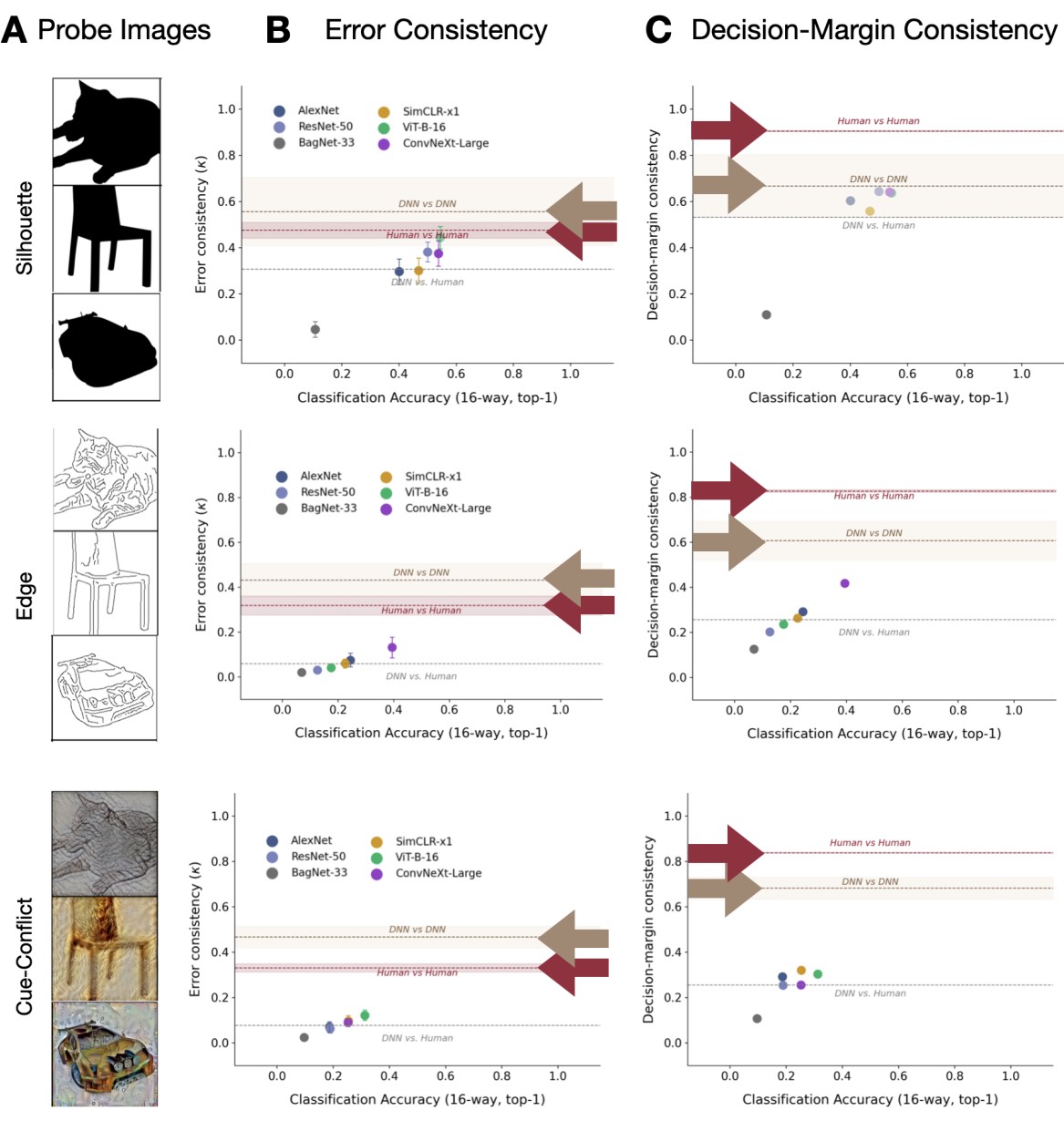

Figure 2: *Error Consistency vs Decision Margin Consistency*. **(A)**. Example probe stimuli from three different experiments, from Geirhos et al., 2020. **(B,C)** Error-Consistency scores (B) and Decision-Margin Consistency scores (C) are plotted for a set of models, as a function of 16-way object classification (top-1), for each of the three experiments. Horizontal bars indicate the reference consistencies for humans-vs-humans (magenta), DNNs-vs-DNNs (tan), and average DNNs-vs-Humans (gray). Bold arrows highlight these consistence levels for comparison across panels Shaded areas reflect 95% confidence intervals.

# 5 Four components of a perceptual decision

In general, accuracy on a perceptual task, on a single trial, with a single stimulus, will depend on at least four factors: (1) the degree to which the perceptual representation disentangles items along the decision axes, (2) how close the decision axes are to optimal given the input perceptual space, (3) the amount of stochastic noise (perceptual noise and decision noise) in the response, (4) any bias the decision-maker has in reporting each response option. Any model of decision-making alignment between systems, whether human or model-based, must account for these four factors.

**Perceptual Representation.** The first factor relates to the degree to which the *featural encoding* of the model is capable of supporting an accurate decision. For example, two distinct stimuli that have a metameric encoding (e.g. [20]) have no chance of being distinguished by a decision making system that maps perceptual representations to decision variables. In the models here, we assess this factor using the correct category logit vs the maximally incorrect logit, as a measure of this fundamental perceptual encoding limit. In humans, as noted, we do not have access to these internal states; but there is empirical evidence that perceptual performance is in fact limited by stable underlying perceptual encoding limits, with consistent item-level variation across a number of distinct visual cognitive tasks from visual working memory to temporal masking to categorization speed, that is linked to the representational similarity structure of the ventral visual cortex (e.g. [21–23]).

**Read-out.** The second factor constraining a perceptual decision relates to the *read-out process* underlying the final discrete decision. Given some (fixed) perceptual encoding, what is the way information is combined to map onto the response options, or decision variables? The models used here have a 1000-way final representation, which Geirhos et al., mapped to the 16 options of the experiment to assess their accuracy. This is a particular formulation of the decision process; an alternative, for example, would be to train a separate 16-way classifier head directly from a frozen feature backbone. This is equivalent to configuring a task-specific read-out head, and could change the exact pattern of results, without changing the perceptual encoding limits. In humans, the task is conveyed to the participant through natural language, which is thought to configure prefrontal and parietal regions; these attention and task-related processes access (and likely influence) perceptual representations in the ventral visual stream [24], in order to control motor systems for the output response. Thus, there are also constraints in how perceptual representations are mapped to decision variables and/or flexible task representations, which will also influence the accuracy of a response on a given trial, independent of the perceptual representation.

**Stochastic Noise.** The third factor constraining a perceptual decision relates to the amount of *stochastic noise* (either in the perceptual encoding or in the read-out process, or both). Of course, in the DNN models tested here there is no noise, so we have direct access to the constraints of both the perceptual encoding and decision process. In humans, we have here followed standard assumptions that noise on any given trial is unrelated to the stimulus, to the perceptual decision difficulty, and to the read-out decision process. Of course, any of these could be false. For example, independent noise processes might fluctuate with attention or overall effort, which might be deployed systematically on more difficult trials. Or noise processes might originate within the perceptual system where some items might be less well captured by the existing features and thus inherently noisier than more prototypical stimuli from a class. As we highlighted in this work, averaging over repetitions is an effective way to remove noise across all sources; in the limit with infinite trials, the estimated decision margin will not be influenced by this factor.

**Response Bias.** The final factor that influences the accuracy of a perceptual decision is related to *response bias*. Given complete uncertainty on a trial (e.g. if only a mask was shown with no preceding stimulus), people are not guaranteed to uniformly guess amongst the n-alternative choices. Instead, they may instead have a response bias (e.g. always the key under the right index finger, or always the item in the 4x4 grid that is closest to the mouse's current position). Here we simply set this factor aside and focused on handling noise, as is standardly done in signal-detection approaches to n-alternative forced choice tasks, purely for simplicity rather than theoretical purposes. However, systematic work has been done in this area that articulates how one would measure response bias and take it into account when assessing underlying accuracy [17].

# 6 Discussion

Here we introduce decision-margin consistency to measure the degree of agreement in perceptual-encoding and decision-mapping algorithms (humans vs. humans, models vs. models, human vs. models), based on the premise that similar perceptual-decision making systems ought to agree on which images are easy to classify and which are difficult to classify. This method provides a principled method for addressing the challenge of inconsistent responses in human decisions. However, it is important to note that this approach, like all approaches to date, makes several simplifying assumptions about the decision process, in models and in humans, that will warrant continued inquiry going forward.

A key question to grapple with is whether the goal is to model the decisions of individual human decision-makers, or whether you are content with modeling the "average human decision maker" as we did in the present study. For either choice, we recommend collecting at least two trials per stimulus or trial-type in all behavioral experiments, which is the minimum that would allow an estimate of what proportion of variance is driven by noise for any analysis (by splitting the data in half and computing self-consistency in scores).

If the goal is to model an individual human decision-maker's decisions, in order to explain true individual differences between people, then many more trials per item will be required to achieve reasonable self-consistency (see Appendix B). And an important caveat for pursing this single-subject route is that introducing repetitions of items also introduces across-trial effects (priming, adaptation, perceptual learning) that might make the repetitions more dependent on each other over time, thus lessening the cost-benefit advantage of averaging trials to remove noise. Luckily, the most dramatic effects of averaging happen in the first 20-50 trials, with diminishing returns after that, so if repetitions are possible in the design, this will help increase the reliability of assessing and comparing two noisy individuals to each other.

Given that the current results suggest a wider gap between models and the average human than previous work on model-vs-human alignment, we believe there are substantial gains to be made by focusing on "the average human decision", targeting the common representational structure shared between humans before investigating more subtle, and difficult to measure individual differences.

The current work makes contact with the rich literature on obtaining accurate estimates of uncertainty in deep neural network classifiers ([25–27]). For example, here, to estimate a model's decision strength, we leveraged the distance to the decision margin, drawing on principles from signal detection theory. The decision margin can be considered a measure of confidence (or uncertainty) in the decision, since longer distances indicate more evidence in favor of the target (relative to non-targets) than shorter distances. However, other metrics for model's decision uncertainty have been explored like [25, 28–31]. An interesting future direction would be to consider post-processing methods to actively re-callibrate a models uncertainty (e.g. see [25]). It is possible calibrated deep nets may be more consistent with each other than non-calibrated deep nets, and may also have increased consistency with humans. However, it is also possible that these models are actually truly different in their underlying representations and thus in the per-item decision difficulties, and so will in fact become more distinct after calibration. More work is needed here to characterize true individual differences in model-to-model comparisons under different calibration settings.

The strength of the signal detection approach lies in its thorough development towards understanding human decision making, and the fact that these insights can be leveraged to better understand the alignment between models and humans, providing a strong framework for understanding the possible sources of discrepancies between any two decision makers.

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

# A    Appendix: Behavioral Experiment Details

The purpose of this experiment was to replicate the Geirhos et al. [2] "edge" experiment, except with multiple repetitions per stimulus to assess self-consistency (i.e., how consistent people are with themselves across multiple responses to identical stimuli) and to quantify the level of internal noise in single trial responses.

**Task**. The task was a 16-alternative-forced-choice task, where observers had to decide which of 16 categories an image should be classified as (airplane, bear, bicycle, bird, boat, bottle, car, cat, chair, clock, dog, elephant, keyboard, knife, oven, truck). We followed the paradigm of Geirhos et al. [2], with a few adjustments to timing and masking for online testing. As depicted in Fig 1, each trial consisted of a 50ms presentation of the stimulus image, followed by a full-contrast pink noise mask (1/f spectral shape) of the same size lasting for 500 ms, followed by a 500ms blank screen, and then by the response screen which was shown until their response. On this screen, the text-label of all 16 categories were arranged in a 4 × 4 grid, and participants had to choose one of 16 categories by clicking on it.

**Stimuli**. Our rule-of-thumb time-budget for online behavioral data collection is to limit testing to approximately 15 minutes, which is just enough time for 160 trials with this paradigm. In the original experiment, participants completed 160 trials (10 images per condition, with no repetitions of the same image), so we divided the images into two sets of 80 (setA and setB), with each image repeated twice. Thus each participant would see 80 unique images (5 per category), each repeated twice during the experiment (160 trials total), and the full set of 160 images would be observed across participants. Stimuli were divided into sets by sorting according to accuracy in the original study, chunking the list into pairs of two, and randomly assigning one of the pair to setA and one to setB. This procedure ensured there was an equal number of images per category for each set, and that the difficulty of the items in each set was well matched, and spanned the full range of difficulty of the full image set.

**Masks**. A given edge image was always paired with exactly the same mask so that variability in classification of the target image could not be due to variability in the mask stimulus.

**Participants**. Participants were recruited via the online behavioral testing platform Prolific. A total of 45 participants completed the experiment for $3.40 ($16.88/hr on average). We applied an exclusion criteria, filtering out any participants that were more than 3 standard deviations from the mean in accuracy averaged over all 160 trials, but there were no outliers by this criterion so no participants were removed from the final analysis.

**Code**. Code for running the generating the masks, the stimulus splits, and for running the online experiment is be available at https://harvard-visionlab/decision-margin-consistency.

# B    Appendix: Using Simulations to Estimate Experiment Power

You can simulate how many trials you need to collect per item using a simple simulation that does not make any assumptions about noise levels. The only assumption is that there is a "true" proportion correct that you would obtain if you had infinite trials to measure accuracy for an item, and that small samples will result in error in estimating that that value (i.e., sampling error). For example, suppose the true percent correct for an item was 75%. To simulate accuracy on N trials, we simply sample from a binomial distribution with probability = .75. The only assumption here is that the probability of responding correctly is the same (.75) on every single trial.

So how do we use this procedure to estimate how many trials we need? We need to define "how close" to the true probability we want to be, then we search for the number of trials that gets us close enough. For example, say we want 95% of our simulations to be within +/-5% of the true accuracy, which is 75% correct for this example. How many trials would we need to achieve that level of precision? The answer is...a lot. Simulating 10000 runs of 10 trials, the 95% confidence interval spans from .5 to 1.00, with 47.31% of simulations yielding estimates outside our +/-5% criterion. This means that if you run an experiment with 10 trials per item, and the true accuracy for an item is .75, there's a 95% probability that your "estimated accuracy" will be somewhere between .5 and 1.00, and 47% probability it will be lower than 70% or higher than 80%.

Doubling the number of trials to 20 buys us little (95% CI: [.55,.90]; 44.61% of simulations exceed our criterion). Not until we have 300 trials can we assume our estimated proportion correct will be

within +/- 5% of true accuracy of .75 (95% CI: [.70,.796]; 4% of simulations outside our criterion range).

Fig. 3 shows the 95% CI across a range from 10 to 1000 trials for three different true accuracy levels (true proportion correct = .5, .75, or. 90). Note that there are diminishing returns beyond 200 trials. Moreover, the number of trials needed will vary based on the true performance level, with generally fewer trials required to estimate more extreme values (either very high or very low accuracies).

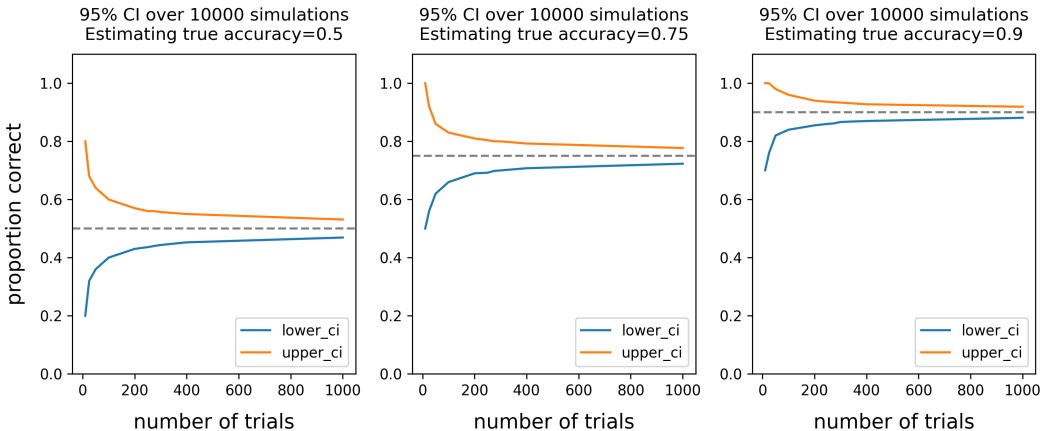

Figure 3: Simulation results, 95% confidence interval for simulations with number of trials ranging from 10 to 1000 when the ground-truth proportion correct is .5, .75, and .90.

## B.1   Decision-Margin Consistency Power Analysis

We conducted a power analysis to estimate the reliability of our human vs. human decision margin consistency measure using the same assumptions as in the previous section. Simulating sampling error only, how consistent will estimated item accuracy correlate with true item accuracy? How many subjects would we need to run to have a correlation > .90 with the ground truth accuracy scores? We find between 20-40 subjects is sufficient to estimate item-level accuracies that correlate with the ground-truth with $r > .90$ (4). The number of subjects required depends on the ground truth distribution of accuracy scores, so we sampled "simulated ground truth" from a distribution of scores that mirrored the observed distribution of item accuracies. Thus, with N=45 participants, we have high power for estimating the variability in decision-margin distance across items (over the range and distribution of observed individual-item accuracies).

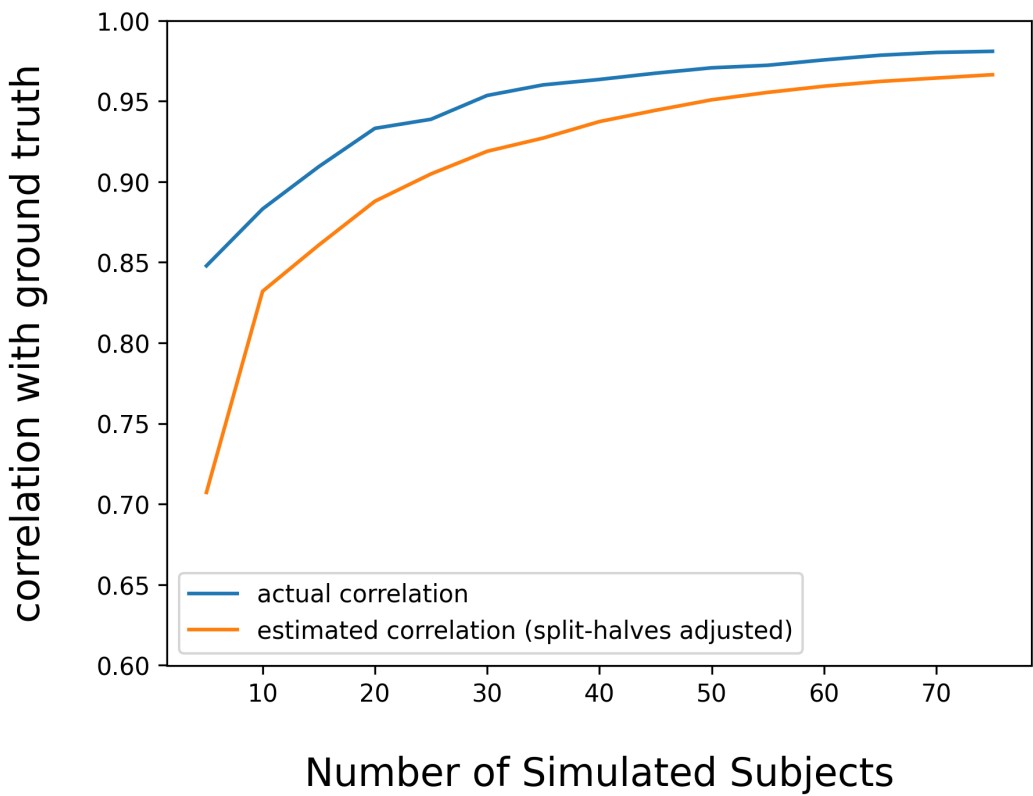

Figure 4: Simulation results, correlation with ground truth for different sample sizes. A sample of 20-40 subjects is sufficient to achieve a high correlation (>.90) with the ground truth, and this correlation with ground-truth can be well-estimated by split-half correlation analysis (though the truth correlation with groundtruth is underestimated slightly).

