# OpenReview forum: "Decision-margin consistency: a principled metric for human and machine performance alignment"
_NeurIPS.cc/2024/Workshop/UniReps — UniReps_

### Official Review · Reviewer_YJtw · 2024-10-03
**Intriguing ideas but execution can be improved**

**Rating:** 5
**Confidence:** 4

**Review:**

The paper introduces the idea of using decision-margin consistency as a measure for human-human and human-machine alignment. They motivate their approach by high variability within subjects, when they solve the same task twice (high noise). By averaging over multiple trials this noise should cancel out, however this would need over 300 trials in this scenario. They present two new metrics, *decision margin* and *decision margin index* to estimate the decision margin in networks and humans respectively, based on logits for networks and simple averaging for humans. The presented results show that by using the decision margin consistency, alignment generally increases, especially the human-human alignment. It also results in a bigger gap between human-human/machine-machine and human-machine alignment than currently thought. A striking result is that human-human alignment is higher than machine-machine alignment, which is usually thought of the other way around in the alignment community, which leads me to question the metric as a reliable measure of alignment (but I could be convinced otherwise). The key-take away message of including several trials in user studies and averaging to eliminate noisy answers seems quite relevant.

**Pros:**
- well-written and easy to follow
- motivation is clear
- intriguing results
- chapter 5 is a nice reflection on influences on the alignment that should be considered
- main take-away and broader discussion is quite relevant

**Cons:**
- not enough trials to assume an accurate estimate (experiment has 45 participants, 300 would be needed according to your simulations)
- accuracy of humans and models is not given, would be needed to interpret the results. Not sure if I interpret Figure 2 correctly, but if the dots are the accuracies of the DNNs, those are quite low (<50%) and would make any alignment analysis questionable?
- error consistency is not introduced and described, only appears in the results
- unsure if the introduced metrics offer a good measure of alignment

**Suggestions for Improvement:**
- extend background to include a more complete overview of alignment measures, it is unclear if the idea of averaging over all human responses is novel and how the new measure compares to existing measures
- larger user study would be needed to get reliable results

This work is of interest for the community, however the paper itself needs some work to improve its significance and the methodology seems not well-developed (e.g. too little sample sizes, poor-performing models). Therefore, I have to give it a borderline reject but I do see potential for a future publication.

---

### Official Review · Reviewer_MEcx · 2024-10-04
**elegant but probably too biased**

**Rating:** 5
**Confidence:** 3

**Review:**

The paper introduces a technique to estimate classification performance alignment between humans and machines. Central to the method is the recognition that while classification is a discrete phenomenon, the evidence leading to a decision is a continuous metric. By focusing on the evidence rather than the final decision, the authors aim to better quantify the alignment between human and machine classification. To estimate this alignment, they propose averaging over trials from various subjects for human data and using activation functions for each class in machine models. This approach is straightforward and elegant.

However, there are several significant limitations to the study:

Simplifying Assumptions about Noise and Response Bias: The method relies on the simplifying assumptions that noise is stimulus-independent and that there is no response bias. Both assumptions are incorrect. Although in the current task—a class selection among 16 classes—the impact of these biases may be minimal, these biases could become substantial in tasks with fewer classes, highly dissimilar classes, or classes associated with cognitive properties like emotions. Ignoring stimulus-dependent noise and response bias can lead to inaccurate estimates of alignment.

Estimation of Evidence by Averaging Across Subjects: The current approach estimates evidence by averaging across subjects. While this is feasible for the task at hand, it would be more informative to obtain evidence at the single-subject and single-trial levels. In psychophysics and psychology experiments, evidence levels are often measured using reaction times, pupil size, heart rate variability, and other physiological measures. Additionally, evidence could potentially be estimated by fitting dynamical systems to longitudinal responses—including both choices and reaction times—which might provide better insights into the underlying evidence and improve the estimation of the noise ceiling.

In summary, while the paper presents an interesting idea, it adds little to the current literature. The method has fundamental limitations that could bias our understanding of human-machine alignment in certain contexts. Addressing these limitations—particularly concerning the assumptions about noise and the approach to evidence estimation—would enhance the contribution and applicability of the proposed technique.

---

### Official Review · Reviewer_7DQ8 · 2024-10-04
**A nice contribution that exposes subtleties in DNN-to-human comparisons**

**Rating:** 8
**Confidence:** 3

**Review:**

The authors propose a new metric to quantify human-to-human, machine-to-machine, and machine-to-human decisions called decision-margin consistency.

The idea is quite simple. In essence, instead of correlating binary decision outcomes (i.e. "correct" or "incorrect" decisions) across systems, the metric correlates a measure of decision confidence/consistency across systems. For deep nets, this is the decision margin. For humans, the experiment is modified to include multiple trials of the same stimulus and the average outcome is recorded (e.g. for two repeated trials: `0`, `0.5`, or `1` for always incorrect, 50% correct, and always correct). They show that humans are more consistent to each other by this measure than what is typically reported. Further, there is a larger gap between deep nets and humans, complicating the narrative that deep nets are well-aligned to human perceptual performance.

I have some suggestions for improving the manuscript:

* It would be easier to read the manuscript if the sections were given more standard titles. For example, section 2 might be called "Background" (rather than "human decisions are noisy"). Section 3 might be called "Method" (rather than "A signal detection framework for..."). Section 4 might be called "Results".

* In section 2, the authors should provide details about how the confidence intervals for self error-consistency, et cetera were computed. I think you might need to be careful about how this is done. There were 45 participants, so there are 45-choose-2 estimates (i.e. 990) of the error consistency. However these are not independent estimates because participant data is re-used across these pairwise comparisons. Off the top of my head, I'm actually not certain how to construct a valid confidence interval for the statistics being reported.

* There is a rich literature on training deep networks to have calibrated estimates of their uncertainty. See for example [Guo et al. (2017)](https://proceedings.mlr.press/v70/guo17a.html). This seems quite related and worthwhile to cite in the introduction / discussion. For example, one would expect that calibrated deep nets may be more consistent with each other than non-calibrated deep nets.

* Likewise, there is a literature on deep networks with stochastic representations that should be cited and discussed. For example, the following seem relevant to neural/psychology audience:
    * [Dapello et al. (2020)](https://proceedings.neurips.cc/paper_files/paper/2020/hash/98b17f068d5d9b7668e19fb8ae470841-Abstract.html)
    * [Duong et al. (2023)](https://openreview.net/forum?id=xjb563TH-GH)

* Figure 2 legend has no explanation for panel C.

* On the top of page 5, there is a reference to equation 4. I believe this is supposed to be to equation 3.

* On the bottom of page 7, there is a typo "influeces" -> "influences"

---

### Decision · Program_Chairs · 2024-10-10

**Decision:**

Accept

**Comment:**

In light of the positive reviewers' feedback and relevancy of the submission, we are pleased to accept this paper for presentation at UniReps 2024. We kindly ask the authors to incorporate the reviewers' suggestions and feedback in the final camera-ready version of the manuscript.